# A Comprehensive Overview of Recent Advances in Epigenetics in Pediatric Acute Lymphoblastic Leukemia

**DOI:** 10.3390/cancers14215384

**Published:** 2022-11-01

**Authors:** Paulina Drożak, Łukasz Bryliński, Joanna Zawitkowska

**Affiliations:** 1Student Scientific Society, Department of Pediatric Hematology, Oncology and Transplantology, Medical University of Lublin, A. Racławickie 1, 20-059 Lublin, Poland; 2Department of Pediatric Hematology, Oncology and Transplantology, Medical University of Lublin, A. Racławickie 1, 20-059 Lublin, Poland

**Keywords:** acute lymphoblastic leukemia, children and adolescents, epigenetics

## Abstract

**Simple Summary:**

Acute lymphoblastic leukemia (ALL) is the most common cancer among pediatric patients. Thanks to the introduction of revolutionary treatment methods and advances in our understanding of the pathogenesis of this disease over the last few decades, survival rates of childhood ALL have increased from less than 10% in the 1960s to almost 90% currently in developed countries. However, there is still a need for further improvement. The latest findings in the field of epigenetics and acute lymphoblastic leukemia show promising potential for the treatment and diagnostics of this disease. The aim of this review is to summarize our current knowledge, according to recent findings, on epigenetics in childhood acute lymphoblastic leukemia, including specific epigenetic alterations in ALL, their potential use as biomarkers for classification, predicting relapse and disease progression, as well as them being targets for novel therapeutic strategies.

**Abstract:**

Recent years have brought a novel insight into our understanding of childhood acute lymphoblastic leukemia (ALL), along with several breakthrough treatment methods. However, multiple aspects of mechanisms behind this disease remain to be elucidated. Evidence suggests that leukemogenesis in ALL is widely influenced by epigenetic modifications. These changes include: DNA hypermethylation, histone modification and miRNA alteration. DNA hypermethylation in promoter regions, which leads to silencing of tumor suppressor genes, is a common epigenetic alteration in ALL. Histone modifications are mainly caused by an increased expression of histone deacetylases. A dysregulation of miRNA results in changes in the expression of their target genes. To date, several hundred genes were identified as suppressed by epigenetic mechanisms in ALL. What is promising is that epigenetic alterations in ALL may be used as potential biomarkers for classification of subtypes, predicting relapse and disease progression and assessing minimal residual disease. Furthermore, since epigenetic lesions are potentially reversible, an activation of epigenetically silenced genes with the use of hypomethylating agents or histone deacetylase inhibitors may be utilized as a therapeutic strategy for ALL. The following review summarizes our current knowledge about epigenetic modifications in ALL and describes potential uses of epigenetics in the clinical management of this disease.

## 1. Introduction

Acute lymphoblastic leukemia (ALL) predominantly affects children (80% of cases). In recent times, the 5-year survival rates of pediatric ALL for children (age 0 to 14 years) and adolescents (age 15 to 19 years) have significantly augmented: from 73% and 55% in the years 1980–1990 to 93% and 74% in the years 2010–2017, respectively [1].

Since the St Jude Total Therapy Study 16 on pediatric ALL patients revealed that the intensity of conventional chemotherapy has reached its limit, there is a need to combine or replace traditional chemotherapy with immunotherapy and molecular targeted therapy [2,3]. The study of epigenetic modifications opens the door for further increasing the survival rates and quality of life of patients with childhood ALL.

Epigenetics is a term used to describe a gene regulation mechanism that takes place without changes in the DNA sequence. Epigenetic dysregulation, alongside genetic lesions, has been well-recognized as a significant factor contributing to the development and progression of many cancers [4,5]. However, as opposed to genetic changes, epigenetic alterations are potentially reversible.

Multiple epigenetic abnormalities, such as DNA hypermethylation, histone modification and miRNA alteration, were identified so far in ALL. Numerous latest studies have demonstrated that epigenetic changes play a significant role in the pathogenesis, treatment outcome and relapse of ALL [3]. This makes epigenetics potentially useful in the clinical management of this disease. With the recent progress in technical advancement and decreasing costs of high-throughput DNA sequencing and Single Nucleotide Polymorphism (SNP) genotyping, multiple novel studies on epigenetic changes in pediatric ALL have emerged [6].

## 2. DNA Methylation in Pediatric ALL

DNA methylation is an epigenetic modification that is performed through an addition of a methyl group to the C5 position of cytosine via enzymes called DNA methyltransferases (DNMTs). This results in a formation of 5-methylcytosine. DNA methylation could occur in the whole genome, but most commonly it occurs in the CpG islands (CGIs), which are regions rich in CpG dinucleotides in mammalian gene promoters [7]. Methylation of CGIs results in a reduction of gene expression [8]. Abnormal CpG methylation patterns have been associated with the development of cancer. An extensive hypermethylation of CGIs located in promoters leads to silencing of related genes, including tumor suppressor genes. Furthermore, widespread hypomethylation of CpGs outside of promoter sequences increases genome instability and thus promotes the emergence of chromosomal abnormalities, which facilitates cancerogenesis [9]. This hypomethylation is thought to be induced by a deficiency in so-called ten-eleven translocation (TET) proteins, a group of enzymes that catalyze oxidation of methylated cytosine [10].

### 2.1. Hypermethylated Denes and Altered Methylation Profiles in Pediatric ALL

Promoter hypermethylation is a crucial attribute of ALL [11]. This was confirmed in a newest study performed by Hetzel S. et al. using whole-genome bisulfite sequencing across ALL subtypes, leukemia cell lines and healthy hematopoietic cells revealed that ALL is significantly characterized by CGIs hypermethylation. However, contrary to most cancer types, ALL samples exhibit minimal global loss of methylation [12].

Many genes have been so far identified as hypermethylated in both pediatric B-ALL and pediatric T-ALL. These genes include TP73 (a member of the p53 gene family); TET2 (crucial for DNA methylation); CDNK1A (p21), CDKN2B (p15) and CDKN2A (p16) (associated with the cell cycle); sFRP2, DKK3 and WIF1 (involved with the WNT pathway), PTEN (a tumor suppressor) and SYK (engaged in the JAK/STAT signalling pathway) [13]. Genome-wide analyses have demonstrated that all genetic subtypes of ALL have a common methylation signature. Furthermore, different, distinct methylation signatures were identified for individual genetic subtypes of ALL. Figueroa et al. in their study comprising samples of 137 B-lineage and 30 T-lineage childhood ALL cases found that different genetic subtypes of ALL exhibited their own specific DNA methylation signatures that correlated with gene expression profiles. Overall, B-ALL and T-ALL varied by 1319 differentially methylated regions (DMRs). Differences were also visible across B-ALL subtypes with the strongest signature observed in high hyperdiploid B-ALL (719 DMRs) and least pronounced signature in CRLF2r B-ALL (52 DMRs). Furthermore, authors also described an epigenetic methylation signature present in all types of ALL, which suggests a core set of genes crucial to the initiation or maintenance of malignant transformation of lymphocytes [14]. A later study conducted by Wong et al. showed that an assessment of methylation in only fifteen loci was sufficient in order to differentiate leukemia from disease-free samples and purified CD34+ cells. A pattern of DNA methylation in these loci was recurrent across different cytogenetic subtypes of pre-B cell ALL [15].

### 2.2. Methylation Patterns as a Tool of Assessing Prognosis of Pediatric T-ALL

A discovery of different methylation patterns and their association with different genetic subtypes of ALL raised a question, whether this knowledge could have a prognostic significance for clinical outcomes of patients. To date, two epigenetic classifiers of prognosis in T-ALL were described: CIMP (CpG Island Methylator Phenotype) and COSMe (CpG island and Open Sea Methylation).

The CIMP classification was to date investigated in multiple studies, both on child samples, as well as on adult samples [16,17,18,19,20]. The first such study conducted on pediatric ALL samples only was conducted by Borssén et al. The authors identified two distinct CIMP groups among pediatric T-cell acute lymphoblastic leukemia (T-ALL) samples based on the most differentially methylated CpG spots. Patients with a positive CGI methylator phenotype (CIMP+) had a superior event-free survival and superior overall survival compared to those without CGI methylator phenotype (CIMP-) [16]. Borssén et al. also investigated the prognostic value of CIMP classification among pediatric B-cell progenitor acute lymphoblastic leukemia (BCP-ALL) patients. The authors analyzed BCP-ALL samples from pediatric patients and found that those with a CIMP- profile at initial diagnosis had a lower overall survival than those with CIMP+ profile [21]. Haider et al. conducted a study that explored the biological basis behind the CIMP subgroups. The CIMP+ subset, characterized by better prognosis, showed indicators of longer replicative history, such as shorter telomere length, older epigenetic age and older mitotic age. Furthermore, the CIMP+ subgroup showed significantly higher expression of ANTP homeobox oncogenes, as well as novel T-ALL genes including Phospholipase C Beta 4 protein coding gene (PLCB4), Plexin D1 protein coding gene (PLXND1), and Myosin XVIIIB protein coding gene (MYO18B). Whereas the CIMP- subgroup, associated with worse prognosis, showed stronger expression of TAL1 with recurrent STIL-TAL1 fusions and higher expression of Brain Expressed X-Linked 1 protein coding gene (BEX1). Overall, the authors suggest that distinct methylation patterns indicate different routes to leukemogenesis in the CIMP+ and CIMP- T-ALL subgroups [22]. The newest study further investigating molecular basis of CIMP by Roels et al. revealed that methylation of CGIs in T-ALL mostly occurs at promoters of Polycomb Repressor Complex 2 (PRC2) target genes, which are not expressed in normal or malignant T cells and that display a reciprocal association with H3K27me3 binding. The authors also discovered that this abnormal methylation profile is a reflection of epigenetic history of T-ALL, since it is already present in pre-leukemic, self-renewing thymocytes that lead to the development of T-ALL [20].

The COSMe (CpG island and Open Sea Methylation) is a classification newly proposed by Roels et al. based on a study conducted on pediatric T-ALL samples. It includes three clusters based upon which each T-ALL could be classified into one of the two groups: COSMe type I (COSMe-I) and COSMe type II (COSMe-II). The COSMe cluster A strongly overlaps with the regions taken into account in the CIMP classification and corresponds mainly with the promoters of PRC2 target genes. These sites are hypomethylated in COSMe-I and hypermethylated in COSMe-II. Moreover, it was discovered that genes controlled by CGIs in cluster A have no or low expression, regardless of their COSMe subtype. It was suggested that in the COSMe-II group those genes are silenced by the hypermethylation of CGIs, whereas in the COSMe-I group, the silencing is caused by H3K27me3 (PRC2-mediated repression), a marker of silent chromatin, and cluster B and C are mostly placed all along genes and intergenic regions. The methylation pattern in cluster B enables us to differentiate between immature and mature T-ALLs. In the case of COSMe-I, cluster B is characterized by high DNA methylation, whereas in the case of COSMe-II, two subgroups with low and high methylation, respectively, could be distinguished, whereas immature T-ALLs show low cluster B methylation. In general, the COSMe-I subgroup was described as epigenetically young, whereas the COSMe-II subgroup was labeled as epigenetically old. The COSMe-I subtype represents a more aggressive form of T-ALL than COSMe-II. This is due to younger mitotic age and faster cellular proliferation in COSMe-I than COSMe-II [23].

The CIMP- subgroup largely corresponds with the COSMe-I subtype, whereas CIMP+ corresponds with the COSMe-II. Overall, the COSMe-I/CIMP- cases are characterized by younger mitotic age (shorter history of proliferation), which is visible in lower methylation (hypomethylation). This might be considered a marker of higher aggressiveness of ALL cells, since less cell divisions preceded the emergence of overt leukemia, which is reflected in the worse prognosis of patients with hypomethylated/COSMe-I/CIMP- subtype of ALL. In turn, the COSMe-II/CIMP+ cases are characterized by older mitotic age (longer history of proliferation), which is reflected in higher methylation (hypermethylation). In cases of COSMe-II/CIMP+/hypermethylated subtype, the aggressiveness of ALL cells seems to be lower, because the latency of the emergence of overt leukemia is longer, which is visible in higher methylation acquired by cells through aging [20,22,24].

### 2.3. Methylation Patterns Combined with Minimal Residual Disease (MRD) as an Improvement of Risk Stratification in Pediatric ALL

The assessment of minimal residual disease (MRD) is a very important prognosticator in pediatric ALL used for risk stratification and the adjustment of therapy [24]. Borssén et al. carried out an analysis of methylation status in pediatric T-ALL samples, divided them according to their CIMP phenotype and examined them in relation to clinical data of patients. The authors found that CIMP classification at diagnosis predicted long-term outcomes of patients who were MRD ≥0.1% at day 29 of therapy (high-risk patients). It was established that the 3-year cumulative incidence of relapse (CIR3y) equaled 50% among CIMP- patients, whereas among CIMP+ patients it equaled 12%. The authors concluded that the CIMP classification at diagnosis added to the MRD assessment could improve risk classification of patients and treatment decision making [19]. Another study by Li et al. in which authors analyzed samples of childhood ALL patients indicated that hypermethylation of CpG sites upstream of Caspase 8 associated protein 2 coding gene (CASP8AP2) was associated with the presence of MRD at day 78. Moreover, patients from the high methylation group had a poor treatment outcome. The authors concluded that combining the methylation level and MRD at day 33 could enhance the risk stratification for ALL among pediatric patients [25].

### 2.4. Hypomethylating Agents in Pediatric ALL

A discovery of the role of hypermethylation in cancer has inspired the development of a novel class of drugs called DNA methyltransferases inhibitors, which act as hypomethylating agents (HMAs) (Figure 1). To date, inhibitors of DNA methyltransferases have been successfully applied and approved by the Food and Drug Administration (FDA) in two other malignancies: acute myeloblastic leukemia (AML) and myelodysplastic syndromes (MDS) [26,27]. The main representatives of this class of drugs are decitabine, azacitidine, zebularine and EPZ-5676 (pinometostat). The first three aforementioned drugs are analogs of nucleosides, which act through the depletion of DNA methyltransferases, and EPZ-5676 is a selective inhibitor of Disruptor of telomeric silencing 1 (DOT1L) [27,28,29]. To date, several pre-clinical and clinical studies involving children were performed in order to evaluate the efficacy of DNMT inhibitors alone or in combination with other drugs in ALL (Table 1).

Overall, DNMT inhibitors appear to be promising in the treatment of pediatric ALL. Decitabine appears to be safe and effective alone and in combination with Hyper-CVAD, after allo-SCT, both in B-cell acute lymphoblastic leukemia (B-ALL) and T-ALL [31,33]. However, conflicting evidence was found in the case of combining decitabine with vorinostat, a histone deacetylase inhibitor. A 2014 study by Burke et al. in which 13 T-ALL and B-ALL pediatric and adult patients were treated with decitabine and vorinostat plus chemotherapy has proven this combination to be tolerable and effective [30]. However, a 2020 study on a larger group of 23 pediatric and young adult patients with B-ALL revealed that a combination of decitabine and vorinostat is clinically unfeasible due to high incidence of significant toxicities [36]. Azacitidine in pediatric ALL is currently under ongoing research on humans, same as pinometostat [32,34,35]. What is promising is that one study on mice models bearing BCP-ALL cells showed that azacitidine overcame resistance to Moxetumomab pasudotox (a chimeric protein composed of an anti-CD22 Fv fused to a portion of Pseudomonas exotoxin A) and greatly improved survival [35]. Zebularine, despite lacking studies on humans, has proven effective alone or with methotrexate in a study conducted by Andrade et al. on childhood B-ALL and T-ALL cell lines. In both cell lines, zebularine induced apoptosis and decreased clonogenic activity and a combination with methotrexate showed a strong synergistic effect. However, the same study also demonstrated that a combination of zebularine with vincristine led to an antagonistic effect [28].

## 3. Modifications of Histones in Pediatric ALL

Epigenetic modifications also apply to histones. Histones are highly conserved proteins that are involved in the packaging of genomic DNA. Together with DNA, histones build a nucleosome—a complex of DNA wrapped on histones [37]. They perform various functions: histones participate in cellular processes, such as: transcription regulation, control of cellular cycle, chromosome stability and repair of DNA damage [38]. Overall, histones can be classified as core histones: H2A, H2B, H3 and H4 and linker histones: H1 and H5 [39]. The core histones are associated with ALL development mainly through aberrations in their post-translational modifications and disturbances of the enzymes regulating these modifications. These modifications are the acetylation and methylation of histones [40,41].

### 3.1. Acetylation of Histones

Histone acetylation and deacetylation are processes that are engaged in the regulation of gene expression. According to the study by Janczar K et al., a loss of global histone H4 acetylation is common in pediatric BCP-ALL. Moreover, preserved histone H4 acetylation is associated with a favorable outcome of the disease: longer relapse-free survival, event-free survival and overall survival [42]. Histone acetylation and deacetylation are also involved in the regulation of Protocadherin 17 protein coding gene (PCDH17). PCDH17 is a tumor suppressor gene in ALL. Repression of histone deacetylation causes upregulation of PCDH17 expression. This suggests that inhibition of histones deacetylation may be a potential pharmaceutical target for ALL with the disfunction of PCDH17 expression treatments [43]. Histone acetylation and deacetylation are catalyzed by enzymes: histone acetyltransferase (HAT) and histone deacetylase (HDAC). Acetylation is linked with an increase of transcriptional activation, while deacetylation is associated with transcriptional deactivation [44]. CREB Binding Protein (CREBBP) is one of HAT. The mutation causing the inactivation of CREBBP occurs in ALL [45]. In the study by Gao C et al., low CREBBP expression is associated with adverse clinical risk factors: high WBC count, T-cell immunophenotype, carrying BCR-ABL and lack of ETV6-RUNX. Additionally, low CREBBP was correlated with a poor prednisone response. Moreover, the CREBBP expression level can be used to predict patient outcomes—the lower expression level was associated with a worse prognosis [46]. Research carried out on ALL cell lines showed that downregulation of CREBBP is associated with resistance to chemotherapy by daunorubicin. Low expression of CREBBP causes inhibition of leukemia cell proliferation, this lead to a decrease in the effectiveness of daunorubicin [47]. Low CREBBP expression is also related to the risk of relapse—CREBBP mutation was present in the clones that survived therapy [48].

### 3.2. HDAC Inhibitors in ALL Treatment

Dysregulation of histone acetylation and deacetylation can be used as a therapeutic target. Inhibition of histone deacetylase seems to be applicable in the treatment of ALL. A mechanism of HDAC Inhibitors (HDACi) is correlated with restoring acetylation homeostasis in cells and reactivating the expression of tumor suppressors, which lead to an induction of cell-cycle arrest, apoptosis, differentiation and inhibition of angiogenesis and metastasis [49]. Givinostat is one of the HDACi which showed potential therapeutic value in ALL treatment. The study by Li Y et al. suggests that givinostat is effective in ALL with Philadelphia chromosome (Ph+) treatment. Givinostat induces apoptosis and inhibits cellular proliferation of Ph + Pre-B ALL cell lines [50]. Givinostat is also useful in the treatment of cytokine receptor-like factor 2 (CRLF2)-rearranged BCP-ALL. This mutation is associated with a poor outcome. In a study from 2017, givinostat exerts cytotoxicity against leukemic cells in the form of growth inhibition and an induction of apoptosis in ALL cells [51]. The research by Yang L et al. showed that a class I and IIb HDAC highly selective inhibitor purinostat mesylate exhibited an antitumor effect on Ph+ B-ALL. The use of purinostat mesylate results in attenuated progression and significantly prolonged survival time in a mouse model [52]. The other HDAC inhibitor panobinostat seems to be effective in the treatment of Mixed Lineage Leukaemia (MLL)-rearranged ALL. In the mouse model panobinostat has strong anti-leukaemic effects, extending survival and reducing the overall disease burden [53]. Another study suggests that panobinostat has better efficiency in combination with the curaxin CBL0137 in MLL-rearranged ALL treatment. These twice anti-cancer drugs destabilize nucleosomes and increase histone acetylation levels. Due to these two processes, CBL0137 and panobinostat decondensed chromatin, respectively, which leads to downstream anti-cancer processes. This combination causes a limit of progression and extends the survival time in the mouse model [54]. Romidepsin also showed efficiency in MLL-rearranged ALL. Research carried out on a mouse model showed that the use of romidepsin significantly enhanced the effect of cytarabine: romidepsin probably affects the cytidine-metabolizing pathway by reducing the expression of cytidine deaminase, an enzyme involved in deactivation of cytarabine, and increases the DNA damage–response to cytarabine [55,56].

### 3.3. Methylation of Histones

Acetylation of histones is not the only histone modification that is involved in ALL development, but also histones methylation is engaged. According to the study by Mei M et al., symmetric dimethylation of H4R3 (H4R3sme2) was upregulated and was due to high expression of protein arginine methyltransferase 5 (PRMT5) in BCP-ALL pediatric patients. Upregulation of H4R3sme2 and PRMT5 occurs in patients with pediatric ALL and is associated with dysregulation of B-cell lineage differentiation. Therefore, an inhibition of PRMT5 expression and its methylation activity may be a potential therapeutic target [57]. Other histones’ methyltransferase plays an important role in the development of ALL. DOT1L is a methyltransferase involved in histone H3 lysine methylation (H3K79me) [58]. This methylation is an important driver of MLL-rearranged ALL, because it is associated with increased transcription of MLL-rearranged ALL genes [59]. For this reason, inhibition of DOT1L can be a therapeutic option in MLL-rearranged ALL treatment, but more research is needed to evaluate the usefulness of DOT1L enzymatic activity inhibitors, DOT1L degraders, protein–protein interaction inhibitors, and combinatorial interventions in MLL-rearranged ALL treatment [60].

## 4. MiRNA in Pediatric ALL

In ALL, expression of some specific microRNAs (miRNA) can be increased or decreased. For this reason, they can be classified as oncomiRs or tumor suppressors: the upregulation of expression suggests that miRNA is oncomiR, whereas downregulation classifies miRNA as tumor suppressor. A distortion of the level of expression of some specific miRNAs can be used in the diagnosis, classification and treatment of ALL [61] [Table 2]. MiRNAs are short non-coding fragments of RNA consisting of 21–22 nucleotides. In normal conditions, by participating in the post-transcriptional regulation of genes, they regulate basic biological processes in the cell, such as growth, proliferation and apoptosis [62]. MiRNA is also involved in the regulation of function and maturation of lymphocytes, regulations of tyrosine kinase and Ras signaling pathways [63].

### 4.1. Epigenetic Regulation of miRNA Expression in ALL

The expression of miRNA can be regulated through hypermethylation and hypomethylation of miRNAs gene promoters. The study from 2009 showed that expression of miR-124a can be regulated by hypermethylation of miR-124a gene promoters: hypermethylation of the promoter is associated with downregulation of miR-124a in ALL cells [83]. Additionally, research by Mi S et al. carried out on 18 bone marrow samples with MLL-rearranged ALL showed that the expression of miR-128 depends on hypomethylation of the CGIs in the promoter region: the expression level of the miRNA has a negative correlation with the degree of methylation of the CGIs [84]. Another study from 2011 suggests that miR 34b expression in MLL-rearranged ALL is downregulated by methylation of CGIs in its gene promoter [85]. In BCP, ALL with TCF3-PBX1 genetic subtype promoters of miR1273G, miR1304 and miR663 were hypermethylated, whereas promoters of miR4442, miR155 and miR3909 where hypomethylated [86]. A study by Faber J et al. suggests that in Ph+ ALL downexpression of miR-203 is induced by promoter CpG hypermethylation [87].

### 4.2. Use of miRNA in Diagnosis and Treatment of ALL

According to a study by Shafik RE et al., miRNA-128 is one of the miRNAs that may be used in the diagnosis of ALL. This research analyzed the miRNA-128 levels in bone marrow cells from 56 newly diagnosed patients with childhood ALL. Overall, 83.9% (*n* = 47) of patients showed overexpression of miRNA-128. Compared to the control group, the level of miRNA-128 in ALL patients was higher [64]. Moreover, the study from 2021 showed that a higher level of miRNA-128-2-5p and miR-378 detected in the blood is associated with ALL relapse, and they can be used as biomarkers for early detection of ALL relapse [65]. The level of miR-181a is also increased in ALL [66]. According to the study by Lyu X et al., miR-181a-5p promotes ALL cell proliferation—suppression of Wnt Inhibitory Factor 1 (WIF1) gene by miR-181a-5p lead to the Wnt pathway activation, resulting in carcinogenesis through dysregulation of cell proliferation and differentiation. For this reason, inhibition of miR-181a-5p can be used as a therapeutic strategy for the treatment of ALL [67]. Moreover, in research by Egyed B et al. carried out on various ALL genetic subgroups as normal karyotype, hyperdiploid, high hyperdiploid, KMT2A-rearranged, P2RY8-CRLF2 translocated, CDKN2A deleted and ETV6 deleted, the level of miR-181a or miR-181a in combination with Vascular endothelial growth factor A (VEGF-A) detected in the cerebrospinal fluid can be useful in the future as a marker of central nervous system (CNS) involvement in pediatric ALL [68,69]. A study by El-Khazragy N et al. found that a high level of miR-181a and miR-155 in bone marrow cells was correlated with a higher risk of recurrence and poor response to treatment. This research also showed that the levels of both miRNAs were downregulated after chemotherapy, which suggest that they can be used as markers of successful therapy [71]. MiR-155 can also promote ALL cell proliferation and inhibit apoptosis, its high level is associated with poor prognosis. Because of that, miR-155 had potential value as a biomarker for predicting prognosis [72]. The study from 2021 suggest that an increased level of circulating MiR-146a in the plasma seems to be a non-invasive predictive biomarker for B-ALL and T-ALL. Monitoring the expression of miR-146a can be used to assess the response to treatment: a reduction of miR-146a expression after chemotherapy seems to be associated with a good response to treatment [73]. Moreover, Anti-miR-146a, Anti-miR-155, and Anti-miR-181a have antileukemic effects. The study by Durmaz B et al. suggests that they affect viability, proliferation, cell cycle and apoptosis of leukemia cell lines and may be useful in therapies for pediatric ALL [70]. A study by Boldrin E et. al suggests that expression of miR-497 and miR-195 is downregulated in BCP-ALL. Moreover, lower expression of both miRNAs is associated with early relapse and shorter relapse-free survival, whereas higher expression of both miRNAs suppressed in vivo growth of leukemia [74]. Lower expression of miR-223 is also characteristic for ALL. Moreover, the level of miR-223 can be used as a possible predictor of ALL relapse and in the monitoring of the treatment: its expression increases in parallel with successful treatment [75]. Because miR-223 inhibits cell proliferation and promotes apoptosis in ALLm upregulation of this miRNA may be a potential therapeutic target [76]. Expression of miR-125b can be decreased in pediatric patients with ALL and is associated with poor disease outcome and malignancy progression. In addition, miR-125b has a regulatory role in B-cell lymphoma 2 (Bcl-2). An expression of miR-125b is related to an expression of Bcl-2: a downregulation of miR-125b is related to Bcl-2 upregulation. The miR-125b Bcl-2 combination can be used to monitor response to therapy [77]. Additionally, the study by Piatopoulou D et al. showed that an expression of miR-125b in bone marrow can be used for monitoring the efficiency of Berlin–Frankfurt–Münster (BFM) therapy. The research was carried out on 125 childhood patients: 88.0% (*n* = 110) of patients had B-ALL, whereas 12.0% T-ALL. 17.6% (*n* = 22) of patients presented high hyperdiploidy (>50 chromosomes) type of ALL, 3,2% (*n* = 4) Hypodiploidy (≤45 chromosomes) type of ALL. Additionally, 1.6% (*n* = 2) were Philadelphia chromosome BCR-ABL1 positive, 24.8% (*n* = 31) had TEL-AML1 (ETV6-RUMX1) fusion gene. A reduced level of miR-125b on diagnosis day and higher 33 day after BFM induction/diagnosis day expression ratio correlate with unfavorable clinicopathological prognostic features: higher risk for disease recurrence and poor survival outcome [78]. A study by Yadav M et al. showed that an evaluation of an expression of circulating miR150 may help in assessing the effectiveness of bone marrow ablation and reconstitution of marrow after transplantation [79]. Expression levels of other miRNAs can be used as diagnostic and multidrug-resistant biomarkers in ALL. MiR-324-3p and miR-508-5p regulate ATP Binding Cassette Subfamily A Member 3 protein coding gene (ABCA3) expression. Overexpression of this gene is associated with increased chemoresistance in ALL. There is a negative correlation between the expression of these two miRNAs and ABCA3 expression. For this reason, an underexpression of these miRNAs suggests multidrug-resistant ALL. Additionally, expression is lower compared to the non-cancer control group [80]. MiRNAs can also influence the effectiveness of glucocorticoids in ALL treatment. Since miR-124 decreased Nuclear Receptor Subfamily 3 Group C Member 1 protein (NR3C) protein expression, which is the receptor for glucocorticoids, expression of miR-124 is significantly higher in children with glucocorticoids-insensitive ALL compared to children with glucocorticoids-sensitive ALL. Moreover, miR-124 is a possible therapeutic target in ALL with glucocorticoids resistance [81]. Whereas a study by Tian C et al. revealed that miR-503 is downregulated in glucocorticoid-resistant leukemic cells and overexpression of miR-503 promotes sensitization to dexamethasone [82].

## 5. Conclusions

Epigenetic dysregulation is known to be an important component contributor to the pathogenesis of cancers. Therefore, increasing evidence on the role of epigenetic alterations in pediatric ALL has inspired novel research in the quest for its potential uses in the management of this disease. So far, knowledge on epigenetics in childhood ALL has opened the door for significant improvements. DNA methylation is an excellent tool for the assessment of prognosis (through CIMP and COSMe classifiers), and combined with MRD could enhance risk stratification. Some specific miRNAs affect the expression of genes involved in the development of leukemia, which makes them a potential therapeutic target. Certain miRNAs could be used as biomarkers of relapse and in the monitoring of the treatment. Furthermore, epigenetic modifications make good targets for therapeutic intervention. A novel class of drugs called DNA methyltransferases inhibitors, which act as hypomethylating agents, have proven effective and safe in studies on humans. Histone deacetylase inhibitors also present therapeutic value, but more studies on humans are needed. Moreover, several types of anti-miRNAs present as candidates for the treatment of ALL.

### Future Perspectives

Future studies should further explore a landscape of epigenetic changes in pediatric ALL, mainly through high-throughput DNA sequencing. This would enable us to increase our understanding on how epigenetic changes influence the clinical phenotype of ALL and further improve risk stratification strategies. Research on drugs targeting epigenetic modifications is particularly important. Pre-clinical and clinical studies should investigate epigenetic drugs in order to develop effective treatment strategies with minimal toxicity. Later, combinations of epigenetic drugs with chemotherapy and/or immunotherapy should be thoroughly researched.

## Figures and Tables

**Figure 1 cancers-14-05384-f001:**
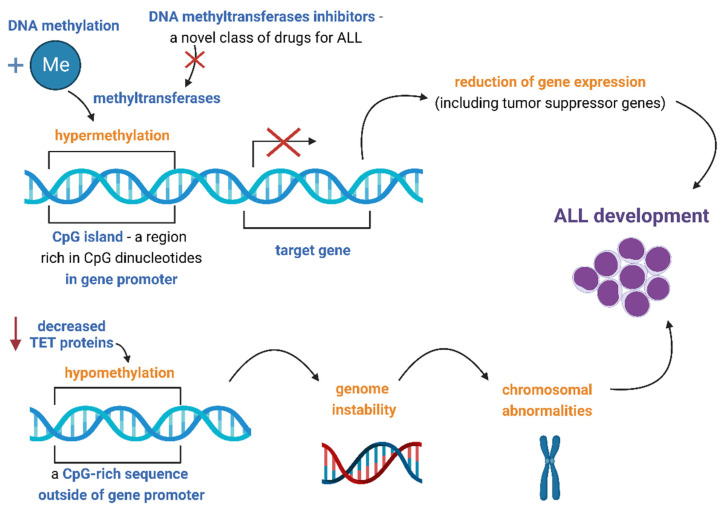
DNA methylation in pediatric ALL. Abbreviations: Me—methyl group; CpG—cytosine and guanine separated by one phosphate group; TET—ten-eleven translocation; ALL—acute lymphoblastic leukemia. Image created with biorender.com.

**Table 1 cancers-14-05384-t001:** A summary of finished and ongoing studies on DNMT inhibitors in pediatric ALL.

Author (Year)/National Clinical Number (Start Year) Additional Information	Hypomethylating Drug Other Drugs	Type of Leukemia	Age of Patients (Number of Patients)/Name of Cell Lines	Results	Grade 3 or 4 Toxic Effects
Andrade et al. (2014) [28]	**Zebularine (ZB)** alone or with methotrexate and vincristine	T-ALL (Jurkat cell line) and BCP-ALL (ReH cell line)	Childhood ALL cell lines (Jurkat and ReH cell lines)	In both cell lines: - ZB decreased clonogenic capacity and induced apoptosis; - combination of ZB with methotrexate resulted in a strong synergistic effect; - combination of ZB with vincristine led to an antagonistic effect.	-
Burke et al. (2014) [30]	**Decitabine,** vorinostat + chemotherapy	relapsed/refractory ALL 1/13 (8%) patient had T-ALL; 12/13 (92%) patients had B-ALL	0–60 years (*n* = 13) 9/13 (69%) patients were children	A significant genome-wide hypomethylation was observed. Decitabine with vorinostat followed by reinduction chemotherapy showed clinical benefit in relapsed patients with ALL and was tolerable.	infection with neutropenia, fever/ neutropenia
Benton et al. (2014) [31]	**Decitabine** alone or with Hyper-CVAD (fractionated cyclophosphamide, vincristine, doxorubicin and dexamethasone alternating with high-dose methotrexate and cytarabine)	32/39 (82%) patients had B-ALL; 7/39 (18%) patients had T-ALL	4–67 years (*n* = 39)	Some patients who had previously progressed on Hyper-CVAD alone achieved a complete response when decitabine was added. Decitabine alone or given with Hyper-CVAD is safe and effective in patients with advanced ALL.	non-life-threatening hepatotoxicity, hyperglycemia
Shukla et al. (2016) [32] Final Report of Phase 1 Study	**Pinometostat** **(EPZ-5676)**	relapsed/refractory MLL-rearranged acute leukemia	3 months-18 years 9/18 (50%) patients had ALL, rest of the patients had AML or mixed phenotype acute leukemia)	Pinometostat was found safe. 40% of patients showed transient reductions in peripheral or bone marrow blasts, however no objective responses were observed.	-
Cui et al. (2017) [33]	Retrospective analysis of relapsed ALL after allogeneic hematopoietic stem cell transplant (alloSCT) who received **decitabine** therapy	3/12 (25%) patients had T-ALL; 9/12 (75%) patients had B-ALL	12–43 years (*n* = 12) 3/12 (25%) of patients were children)	10/12 (83%) patients achieved complete remission. 1/12 (8%) achieved a partial remission. 1/12 (8%) had no response.	no side effects were observed
Sun et al. (2018) [34] Phase 1 study	**Azicitidine** + chemotherapy	not specified	Children—age not specified (*n* = 2)	Neither of the 2 patients with ALL responded.	not specified
Müller et al. (2018) [35]	**Azacitidine**, Moxetumomab pasudotox	BCP-ALL	NSG mice bearing KOPN-8 or Reh cells	Resistance to Moxe was prevented through a combination with azacitidine. Survival greatly improved in both KOPN-8 and Reh models.	-
Burke et al. (2020) [36]	**Decitabine**, Vorinostat	B-ALL	1–21 years (*n* = 23)	9/23 (39%) subjects achieved a complete response. 5/23 (22%) subjects had stable disease. 9/23 subjects were not evaluable for response due to treatment-related toxicities.	hypokalemia, anemia, febrile neutropenia, hypophosphatemia, leukopenia, hyperbilirubinemia, thrombocytopenia, neutropenia, hypocalcemia, infections

Abbreviations: B-ALL—B-cell acute lymphoblastic leukemia; BCP-ALL—B-cell progenitor acute lymphoblastic leukemia; Hyper-CVAD—Hyperfractionated cyclophosphamide, vincristine, doxorubicin, and dexamethasone; KMT2A—lysine methyltransferase 2A protein coding gene; MLL—Mixed Lineage Leukaemia; T-ALL—T-cell acute lymphoblastic leukemia.

**Table 2 cancers-14-05384-t002:** Summary of the usefulness of miRNA in diagnostic and treatment of ALL.

Author (Year)	MiRNA	Upregulated/ Downregulated	Usefulness
Shafik et al. (2018) [64] Bhatia et al. (2021) [65]	miR-128	Upregulated	Biomarker for early detection of relapse ALL.
Shafik et al. (2020) [66] Lyu et al. (2017) [67] Egyed et al. (2020) [68] Egyed et al. (2022) [69] Durmaz et al. (2021) [70]	miR-181a	Upregulated	Inhibition of miR-181a seems to be a therapeutic target; detecting miR-181a in CSF in future can be used as a marker of CNS involvement; overexpression in bone marrow cells is associated with a higher risk of recurrence and poor response to treatment; useful in monitoring it, anti-miR-181a may be useful in therapies for childhood ALL.
El-Khazragy et al. (2019) [71] Liang et al. (2021) [72] Durmaz et al. (2021) [70]	miR-155	Upregulated	Has potential value as a biomarker for predicting the prognosis and marker of successful therapy; anti-miR-155 may be useful in therapies for childhood ALL.
Shahid et al. (2021) [73] Durmaz et al. (2021) [70]	miR-146a	Upregulated	Can be used to evaluate the response to treatment, anti-miR-146a may be useful in therapies for childhood ALL.
Boldrin E et al. [74]	miR-497 and miR-195	Downregulated	Lower expression is associated with early relapse and shorter relapse-free survival.
Nemes et al. (2015) [75] Li et al. (2020) [76]	miR-223	Downregulated	Can be used as a predictor of ALL relapse and in the monitoring of the treatment; potential therapeutic target.
El-Khazragy et al. (2018) [77] Piatopoulou et al. (2017) [78]	miR-125b	Downregulated	Can be used for monitoring response to BFM therapy.
Yadav et al. (2022) [79]	miR150	-	Expression may be helpful in the estimate the efficiency of blood marrow ablation and reconstitution of marrow after transplantation.
Zamani et al. (2021) [80]	miR-324-3p and miR-508-5p	Downregulated	Underexpression of these miRNAs suggest multidrug-resistant ALL.
Liang et al. (2017) [81]	miR-124	-	Expression of miR-124 is significantly higher in children with glucocorticoids insensitive ALL.
Tian et al. (2021) [82]	miR-503	-	Overexpression promotes sensitized for Dexamethasone.

Abbreviations: ALL—Acute lymphoblastic leukemia; BFM therapy—Berlin-Frankfurt-Münster therapy; CNS—Central nervous system; CSF—Cerebrospinal fluid.

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
