# Peer review of "A Comprehensive Overview of Recent Advances in Epigenetics in Pediatric Acute Lymphoblastic Leukemia"

_cancers, 2022, doi:10.3390/cancers14215384_

Round 1

Reviewer 1 Report

The work is well written and addresses several aspects of the epigenetics of pediatric ALL.

The manuscript is clear and reports numerous significant studies present in the literature.

However, I suggest some additions and modifications.

In particular, in chapter 4 "MiRNA in pediatric ALL", the authors can comment on an alternative but possible mechanism of MiRNA regulation through epigenetics. Indeed, also MiRNAs themselves can be regulated by methylation and this can be an interesting point of view proposed in literature (e.g., in publication PMID: 19435910). Moreover, also additional recent findings on MiRNAs can be included in the review, such as the work of Meyer et al. (PMID: 34098582), in which they describe the MicroRNA-497/195.

Moreover, in Conclusions paragraph, a brief comment on the epigenetics of cancer in general can help to have a broader perspective on this phenomenon and its mechanisms in tumor pathogenesis, suggesting the importance of these phenomena.

In addition, as a minor observation related to the introduction, I suggest changing the order of the paragraphs, putting the text part from "Acute lymphoblastic leukemia... patients with childhood ALL" at the beginning and the second part the paragraphs from "Epigenetics is a term used... in pediatric ALL emerged".

Additional references

PMID: 19435910

Cancer Res. 2009 May 15;69(10):4443-53. doi: 10.1158/0008-5472.CAN-08-4025.

Epigenetic silencing of the tumor suppressor microRNA Hsa-miR-124a regulates CDK6 expression and confers a poor prognosis in acute lymphoblastic leukemia. Xabier Agirre et al.

PMID: 34098582

Blood. 2021 Nov 18;138(20):1953-1965. doi: 10.1182/blood.2020007591.

MicroRNA-497/195 is tumor suppressive and cooperates with CDKN2A/B in pediatric acute lymphoblastic leukemia. Elena Boldrin et al.

Reviewer 2 Report

This manuscript from Drozak and colleagues is a review on recent advances in our knowledge of the epigenetics of acute lymphoblastic leukemia (ALL).  The review covers methylation patterns and hypomethylating treatments, along with histone modifications and HDAC inhibitors and a section on microRNA.  

The review is quite comprehensive, but is often difficult to read in places, and sometimes appears like a list of references rather than a true discussion of the advances.

General comments:

  1. Figure 1:  How does hypomethylation (outside of a promoter) occur and how does it contribute to leukemogenesis? For ALL overall is this hypermethylation or minimal loss of methylation? In 2.1 (line 85 and 86) how do individual subtypes of ALL have both common hypermethylation signatures AND distinct hypermethylation signatures? This needs better description.

  1. Section 2.2 should be evened out by expanding discussion of BCP-ALL and reducing that of T-ALL (or the subheading changed to reflect the actual main content).

  1. Both tables have too much information!  Also if the phase II study is ongoing should this still be part of the review, having no outcome yet?

  1. The introduction to section 3 is very confusing and needs good editing.

  1. Section 4.1 really only discusses miRNA as biomarkers of diagnosis and treatment. Reference should also be made to their potential use in treatment (this is commented on in the conclusion without reference - which therapies?).

  1. Conclusion: What sort of epigenetic changes will be discovered by large scale sequencing?

Reviewer 3 Report

1. Nice job on this comprehensive review of the published data in this subject, however the DNA methylation was mainly indicated studies in T-ALL amd did not talk about correlation with the most common murations that are known for T-ALL including NOTCH1 and PHf6, etc.

2.  The miRNA reviewed in only BCR-ABL1 with mention of ETV6-RUMX1 , however B-ALL includes many subtypes such as hypo or hyper diploid, Ph-like ALL, MLL and TCF3 rearranged. 

The pattern of methylation of miRNA was not discussed for subtypes of B-ALL

When most of these data only done in research setting and not CLIA laboratories then how can we conclude this method be diagnostically helpful?
